# Updated Epidemiology of Gastric Cancer in Asia: Decreased Incidence but Still a Big Challenge

**DOI:** 10.3390/cancers15092639

**Published:** 2023-05-06

**Authors:** Wing Sum Shin, Fuda Xie, Bonan Chen, Peiyao Yu, Jun Yu, Ka Fai To, Wei Kang

**Affiliations:** 1Department of Anatomical and Cellular Pathology, State Key Laboratory of Translational Oncology, Prince of Wales Hospital, The Chinese University of Hong Kong, Hong Kong 999077, China; wsshin@link.cuhk.edu.hk (W.S.S.); xiefuda@link.cuhk.edu.hk (F.X.); 1155147752@link.cuhk.edu.hk (B.C.); kfto@cuhk.edu.hk (K.F.T.); 2State Key Laboratory of Digestive Disease, Institute of Digestive Disease, The Chinese University of Hong Kong, Hong Kong 999077, China; junyu@cuhk.edu.hk; 3CUHK—Shenzhen Research Institute, The Chinese University of Hong Kong, Shenzhen 518000, China; 4Department of Pathology, School of Basic Medical Sciences, Southern Medical University, Guangzhou 510515, China; gonlando@i.smu.edu.cn; 5Department of Medicine and Therapeutics, The Chinese University of Hong Kong, Hong Kong 999077, China

**Keywords:** gastric cancer, *H. pylori*, incidence, mortality, targeted therapy

## Abstract

**Simple Summary:**

Gastric cancer (GC) is regarded as one of the most perilous malignancies globally, with over one billion new cases and seven hundred and eighty-three thousand deaths reported in 2020. The incidence of GC is particularly high in Asian countries. Moreover, multiple oncogenic signaling pathways are activated and implicated in gastric carcinogenesis, leading to malignant phenotype acquisition. This review outlines the most updated epidemiology of GC in Asian countries, along with targeted therapies for GC treatment.

**Abstract:**

Despite the decline in incidence and mortality rates, gastric cancer (GC) is the fifth leading cause of cancer deaths worldwide. The incidence and mortality of GC are exceptionally high in Asia due to high *H. pylori* infection, dietary habits, smoking behaviors, and heavy alcohol consumption. In Asia, males are more susceptible to developing GC than females. Variations in *H. pylori* strains and prevalence rates may contribute to the differences in incidence and mortality rates across Asian countries. Large-scale *H. pylori* eradication was one of the effective ways to reduce GC incidences. Treatment methods and clinical trials have evolved, but the 5-year survival rate of advanced GC is still low. Efforts should be put towards large-scale screening and early diagnosis, precision medicine, and deep mechanism studies on the interplay of GC cells and microenvironments for dealing with peritoneal metastasis and prolonging patients’ survival.

## 1. Introduction

Despite the decline in incidence and mortality rates, gastric cancer (GC) remains the fifth most common cancer globally. GC caused approximately seven hundred and seventy thousand deaths in 2020 [1,2,3]. Asian countries generally have exceptionally higher incidence and mortality rates than other countries. Over 60% of GC has recently been reported in Eastern Asia [4]. Common risk factors for GC include *Helicobacter pylori* (*H. pylori*) infection, genetic alterations, race, diet, and lifestyle [5,6]. Specific inherited GC syndromes, such as hereditary diffuse gastric cancer (HDGC) caused by inactivating mutations in the tumor suppressor gene *CDH1*, may also lead to a higher risk of GC. It is estimated that 80% of patients with *CDH1* mutation will develop GC [5].

Classification of GC is essential for improved diagnosis and treatment. GC is commonly classified in histology as intestinal, diffuse [7,8], and mixed type [8]. While the intestinal type is the most common, specific populations are more prone to diffuse types of GC [8]. The World Health Organization (WHO) has recently defined specific cancers based on their molecular phenotype and histological characteristics [9]. In some rare gastric tumors, specific driver mutations have been identified, such as the characteristic MALAT1–GLI1 fusion gene in gastro-blastoma and EWSR1 fusions in clear gastrointestinal cell sarcoma and malignant gastrointestinal neuroectodermal tumors [9].

In general, the molecular mechanism of GC involves the molecular pathogenesis from normal epithelia to early cancer and, finally, to advanced cancer and invasion metastasis [10]. Different histological types of GC may differ in genetic alterations. For example, Kirsten rat sarcoma viral oncogene homolog (*KRAS*) mutations are more likely to occur in the intestinal type of GC [10]. The molecular features may further classify GC into different types, such as the Asian Cancer Research Group (ACRG) and The Cancer Genome Atlas Consortium (TCGA) [11]. The ACRG molecular classification adopted immunohistochemistry, while TCGA molecular classification adopted the next-generation sequencing technology [11].

The advances in diagnosis in certain developed countries has reduced GC incidences. For instance, in Japan, the Health Law for the Aged has extended GC screening across the country since 1983. Moreover, in recent times, endoscopic examinations have been increasingly used in Japan as a screening method for GC. Similarly, in the Republic of Korea, radiographic screening for GC has been used since 2000 [12].

Recently, new clinical treatments specific to different kinds or stages of GC have been developed, improving the survival of GC patients. Endoscopic submucosal dissection (ESD), which offers a faster recovery rate and involves lower costs, is used to treat patients with early GC rather than surgery [13]. Patients with human epidermal growth factor-2(HER2)-positive GC could receive Trastuzumab in first-line chemotherapy to better treat GC and increase life expectancy [14]. Apart from the above clinical treatments, advances in immune checkpoint inhibitors (ICI) have also been made in recent times. Innovative treatment methods that inhibits the programmed death (PD-1)/ programmed death-ligand 1 (PD-L1) axis with ICI are generally accepted to be an effective way to treat advanced GC in the future [15].

Meanwhile, with a high expression mRNA level and the exposure of epitopes in malignant transformation, Claudin 18.2 (CLDN18.2) was identified as a promising therapeutic target for GC treatment. Clinical trials of Claudin 18.2-targeted monoclonal antibody are in progress [16]. Hence, it would be an ideal candidate for monoclonal antibody binding with the capability of reducing off-target effects.

Scientists and clinicians have developed new platforms for GC research and drug screening. One possible approach is to examine genetics in gastric stem cells in organoid models [17]. Patient-derived xenografts (PDX), a novel platform for translational cancer research, could help to investigate the major molecular features of tumors [18].

## 2. GC in Asia

GC is notably more prevalent in Asian countries compared to other regions. As shown in Figure 1, the International Agency for Research on Cancer of the World Health Organization (https://gco.iarc.fr/; accessed on 3 April 2023) reported that the age-standardized incidence rate of GC per 100,000 in Eastern Asia was approximately two times that of other regions, such as Europe (Figure 1). The high prevalence of GC in Eastern Asia may be due to *H. pylori* seroprevalence rate, *H. pylori* oncogenic genes, dietary habits, and tobacco smoking.

### 2.1. H. pylori Infection

Infection with *H. pylori*, a class I carcinogen, is one of the most critical risk factors for GC [19,20]. From 2000 onwards, the prevalence of *H. pylori* has reduced in Europe but not in Asia [21].

Globally, *H. pylori* infection rate for males was 46.3%, while for females it was 42.7% [22]. Asian countries have a higher seroprevalence rate of *H. pylori* than others, meaning more people in Asia have contracted *H. pylori* before. For example, Republic of Korea in Asia recorded a seroprevalence rate of *H. pylori* at 59.6% [23]. It was only 32.5% for the United States of America [24], a non-Asian country.

Several studies have shown that *H. pylori* eradication may significantly decrease incidence rates in Asian countries. A study conducted two randomized, placebo-controlled factorial-design intervention trials in a high-risk GC county in China. The study reveals that *H. pylori* eradication could decrease the likelihood of precancerous gastric lesions and lower the risk of developing GC [25]. Another study also suggested that *H. pylori* testing and eradication in East and South Asia resulted in 4,966,115 mean disability-adjusted life years gained [26].

A relatively large-scale *H. pylori* eradication trial has also supported that mass eradication could dramatically reduce GC without known detrimental consequences. On Matsu Islands in Taiwan, a place with a high risk of *H. pylori* infection, an extensive eradication was conducted between 2004 and 2018. The coverage percentage was 85.5% after six rounds of comprehensive scanning and eradication. Subsequently, *H. pylori* prevalence decreased from 64.2% to 15.0%. This corresponds to 53% effectiveness in reducing GC incidences compared with 1995 to 2003 [27].

Despite the dramatic reduction in GC incidences achieved through eradicating *H. pylori*, a large proportion of people infected with it are asymptotic [28]. Only 1–2% of the infected people develop gastric adenocarcinoma [29]. There is a high prevalence rate for *H. pylori* in Africa but a low GC incidence rate [30]. This gap may be because *H. pylori* contain different strain that may affect developing complications in GC.

The *H. pylori* genome encodes approximately 500 to 600 strain-specific genes, leading to different diseases [31]. These strains have been classified into highly virulent type I, intermediate, and reduced virulent type II strains [32]. One of the most important virulence factors is CagA, encoded by the CagA gene. Because of the connection between the bacteria’s adhesins on the surface and receptors of the bacterial portion on the host cells, CagA can adhere to the surface of gastric epithelial cells [33] and enter the cytoplasm of cells via the type IV secretion system [34,35]. Such an entry marks the first stage of CagA-caused diseases [36]. After penetrating the host epithelial cells, Src and Abl family kinases tyrosine-phosphorylate to some CagA molecules within several repeating Glu-Pro-Ile-Tyr-Ala (EPIYA) motifs [35,37]. Both phosphorylated and non-phosphorylated CagA molecules were known to bind to host proteins and initiate pathogenic actions. Phosphorylated CagA interacts with the SH2 domain, which causes infected host cells in vitro to undergo actin-cytoskeletal rearrangements, dispersion, and elongation due to *H. pylori* [38]. Unphosphorylated CagA may affect the tight junction of stomach cells [39]. In addition, CagA may inhibit autophagy [40] and activate NF-κB [41]. CagA also may elevate cytokine, e.g., IL8 production that promotes inflammation [40,41]. Such inflammation plays a crucial role in developing GC [42].

Several studies have shown that certain oncogenic strains of *H. pylori* may be more commonly found in Asia [25]. For example, Cytotoxin-associated gene A (CagA), a common virulence factor of *H. pylori*, is dominant in China [25,43].

### 2.2. Dietary Habits

Asian people typically prefer traditional processed, spicy, or smoked food. The Japanese make traditionally processed food, such as salted fish [44]. Traditional Chinese cuisines, such as sausage in chuan flavor and Chinese bacon, are usually smoked [45]. Koreans prefer salty and spicy food due to traditional factors [46]. These may increase the risk for the aforementioned *H. Pylori* infection and increase GC risk or promote the progress of GC into a more advanced stage [19,46,47,48], leading to a higher prevalence of GC in Asia.

One reason behind the correlation between high salt intake and GC is that salt could impede the stomach’s mucosal barrier, causing inflammation [6,46]. In an infected model with *H. pylori* and high salt intake, mice developed gastric tumorigenesis in less than a year [47]. Humans also exhibit a similar pattern. Meta-cohort studies conducted in the Republic of Korea show that people who eat more salty food have a higher risk of developing non-cardia GC [49]. In recent years, an Asian endoscopy investigation also found that people who consume more salt may be more likely to develop atrophic gastritis with intestinal metaplasia [50].

Smoked meat also imposes a higher risk for GC by forming polycyclic aromatic hydrocarbons [50]. A meta-analysis of 11 studies has shown that regular consumption of smoked meat raised the risk for GC by 22% [48]. Carcinogenic capsaicin inside spicy food may cause mucosal damage and contribute to a higher risk for GC [46].

### 2.3. Smoking Behaviors

According to Statista, the three countries with the most significant number of smokers worldwide (i.e., China, India, and Indonesia) are in Asia. Tobacco smoking is positively correlated with GC [48,51]. Smoke contains various carcinogens, including N-nitroso-compounds, that may increase GC risk [52]. Another major substance is nicotine in tobacco, which may promote the proliferation and migration of GC cells by releasing prostaglandin E2, COX-2, VEGF, and ERK. It may also activate ERK and a COX-2-dependent rise in VEGF and VEGF receptors [53].

Meta-analysis of a cohort study revealed that compared to men who never smoked, the risk is elevated by approximately 60% in men who smoke and about 20% in women who smoke [49]. Another project, the “Stomach Cancer Pooling Project”, illustrates that the risk of GC increases with the number of cigarettes smoked daily and the duration of smoking [54].

### 2.4. Alcohol Consumption

Heavy alcohol consumption is generally positively associated with the risk of GC. Specifically, people consuming over 45 g of alcohol daily are usually considered to be drinking alcohol heavily and are more at risk for GC [55]. Alcohol dehydrogenase metabolizes ethanol of alcohol into acetaldehyde, which may cause persistent damage to DNA strands and hinder DNA repair processes [56]. Regarding light alcohol consumption, some studies argued that individuals who consume alcohol to an average amount (e.g., equal to or less than four drinks) have a similar risk for GC to people who do not drink at all [57,58]. In contrast, others consider that alcohol consumption would increase the risk of GC [59].

## 3. The Incidence of GC in Different Asia Regions

While GC is especially prevalent in Asia, it varies geographically in terms of prevention and the current epidemiological situation within the continent (https://gco.iarc.fr/; accessed on 3 April 2023). High GC-incidence Asian regions include China, Japan, Republic of Korea, Vietnam, and Mongolia, whereas Hong Kong and Malaysia have a lower GC incidence (Figure 2).

### 3.1. China

From the data in the International Agency for Research on Cancer of the World Health Organization, it can be clearly discerned that GC is substantially prevalent in China. China has an age-standardized incidence rate of GC per 100,000 cases at 20.6 in 2020. The mortality rate in China is also high. The estimated GC mortality (per 100,000) for all ages in GLOBOCAN 2020 was 15.9 in China, with males being 22.8 and females 9.5. The mortality rate is significantly higher than European countries (5.5) and the United States of America (1.7).

Over the last three decades, China’s disability-adjusted life years (DALY) of GC increased by 19.11%, indicating a vast GC disease burden in China. In 2019, the DALYs in China were 44.21% of total global GC incidences [60]. Yet, age-standardized DALY has declined over these years. In particular, the females’ drop is more significant than the males’ [61].

*H. pylori* prevalence varies across regions in China. The lowest prevalence rate for *H. pylori* was in North China, while Northwest China had the highest [62]. It is to be noted that due to the availability of limited resources, China has mainly focused on investing in urban areas over the past decades. Hence, fewer resources are available for rural areas, leaving the regions underdeveloped [63]. As seen in Figure 3, the gap in mortality rate between rural areas and urban areas of China narrowed over time but remained significant, suggested by Medical Knowledge Service System of China. The Xianyou County, a rural area in the Fujian Province of China [64], has a higher GC mortality rate than the rest of China; its GC mortality rate is about two times higher than the average rate in all of China [65].

Since 2005, cancer screening programs have been conducted in China, which may have reduced its GC incidences in recent years. An upper gastrointestinal cancer screening program was initiated in 2005 for rural regions with high incidences. The program now covers most provinces. Other programs, e.g., Cancer Early Diagnosis Program and the Early Treatment Program in Huai River Basin (2009), were also established in light of high GC incidences [66].

### 3.2. Japan

Japan is one of the areas with the highest probability of GC [52]. However, as suggested by Cancer Information Service, National Cancer Center (https://ganjoho.jp/; accessed on 8 January 2023) and shown in Figure 4, the age-specific incidence rate in Japan declined gradually from 1975 to 2019 [67]. In 2020, following Mongolia, the overall GC rate in Japan remained the second highest, with an age-standardized rate of 31.6 (Figure 5). It also accounted for approximately 12.7% of all GC incidences worldwide. The International Agency for Research on Cancer of the World Health Organization reported that males are more prone to develop GC than females in Japan. In 2020, the age-standardized incidence rate of GC in men was 48.1 per 100,000 cases, while in women it was 17.3 per 100,000 cases.

Risk factors may include high-salt dietary habits and behaviors. Men with BMI ≥ 27 kg/m^2^ are also more susceptible to GC [68]. Another primary reason may be the relatively high incidence rate of *H. pylori* in Japan, a common risk factor for GC. As shown in Figure 6, the prevalence rate of *H. pylori* in Japan is 51.7% [69]. However, due to better hygiene, fewer Japanese people born after 1950 are infected with *H. pylori*. *H. pylori*’s prevalence rate in children is below 10%. Hence, the decreasing trend of GC incidences may continue [4].

GC’s mortality rate in Japan has been declining over time. Despite the high incidence rate, its mortality rate ranks 31st in the world [70]. Since 1983, Japan has extensively conducted GC screening and constantly revised guidelines [12]. Early identification through screening reduced the mortality rate by 30–65% [71]. Japan’s National Health Insurance system was the first in the world to include *H. pylori* eradication for chronic gastritis, which certainly reduced the overall GC mortality rate [72]. Due to early and improved detection and treatment, the mortality rate of GC in Japan decreased significantly from above 70 per 100,000 people in 1985 to around 10 per 100,000 people in 2018 [4]. Nevertheless, despite the declining mortality rate, GC remains one of the leading causes of cancer death in Japan. It is also the second most prominent cause of cancer death for men and the fourth most for women [73].

### 3.3. Republic of Korea

According to the International Agency for Research on Cancer, World Health Organization, GC incidence in Republic of Korea is very high. In Republic of Korea, the age-standardized incidence rate per 100,000 cases of GC in men was 39.7, while in women was 17.6 in 2020. The data from the Korean Statistical Information Service (https://kosis.kr/; accessed on 3 April 2023), as shown in Figure 7, highlights that the trend of age-standardized rates of GC remained relatively stable. Unlike Japan, it did not show a decreasing trend. Similar to other regions, aged individuals are more susceptible to developing GC. In particular, men aged 50 or above are approximately two times more likely to be diagnosed with GC. At the same time, people at younger ages have no significant difference in incidence between sexes, as seen in Figure 8.

Generally, aged people would be more susceptible to developing intestinal-type GC, however, males typically develop this type of GC approximately 10 to 20 years earlier than females. Younger people generally acquire diffuse and mixed types of GC [74]. They are significantly more prone than other populations to develop poorly differentiated carcinoma [75].

Cancer screening in the Republic of Korea for individuals aged 40 and older has been implemented every two years, with a nearly 50% participation rate. Almost 54 million stomach cancer screenings were carried out from 2007 to 2016 [76]. This may have contributed to the increased early diagnosis of GC cases and an improved survival rate in 5 years in 2012–2016 [77]. A study compared people who participated in the national cancer screening program for GC at least once and those who did not. The results showed that those who participated had significantly lower medical care costs in 5 years and better prognoses for 5 and 9 years [78]. The screening program may contribute to increased survival and lowered mortality rates in the Republic of Korea. The 5-year relative survival rate for patients diagnosed with GC increased from 55.7% in 1999 to 2005 to 77.0% in 2013–2019 [79]. The aged-standardized mortality rate decreased from 29.0 per person in 2000 to 7.9 per person in 2020 [80].

### 3.4. Hong Kong

Situated in Asia, Hong Kong, a special administrative region of China, has an intermediate risk of GC [81]. Based on the report of the Hong Kong Cancer Registry, Hospital Authority (https://www3.ha.org.hk/cancereg/; accessed on 3 April 2023), in 2020, the age-standardized incidence rate of GC in men was 9.3 per 100,000 cases, while in women was 6.2 per 100,000 cases. The incidence rate of GC significantly increases with age, and the gap in the incidence rate between males and females widens with age, as shown in Figure 9. The highest rate was seen in individuals aged 85 or above; it was 123.3 for males and 65.8 for females, with a male-to-female ratio of approximately 1.87.

Similar to the Republic of Korea and Japan, it is relatively common to be infected by *H. pylori* in Hong Kong; it has a prevalence rate of 58.4% [69,82]. However, according to the International Agency for Research on Cancer, World Health Organization, and the Hong Kong Cancer Registry, Hospital Authority, Hong Kong has a lower incidence rate of GC than these countries. This may be due to the genotypic variations in the VacA gene in *H. pylori* strains. VacA genes may differ in the signal area (s1 and s2) and the middle area (m1 and m2). GC patients often have s1/m1-type strains. Hong Kong has a higher prevalence of *H. pylori* with the m2 type, while the Republic of Korea and Japan have a higher prevalence of *H. pylori* with the m1 type. This may account for the lower incidence of GC in Hong Kong [81]. In addition, in Hong Kong, *H. pylori* eradication treatment may reduce the risk of developing GC in older individuals (i.e., 60 or above) but not younger individuals [83].

According to the Hong Kong Cancer Registry, Hospital Authority (https://www3.ha.org.hk/cancereg/; accessed on 3 April 2023), the age-standardized incidence rate of GC declined gradually over time. It decreased from 19.9 in 1983 to 8 in 2020, with a percentage decrease of 59.8%, as shown in Figure 10.

### 3.5. Vietnam

Vietnam is a lower-middle-income country in Asia with relatively inadequate healthcare resources. It resulted in a high incidence and mortality rate of GC, with approximately 18,000 new cases and 15,000 deaths in 2018 [84]. According to the International Agency for Research on Cancer, World Health Organization, in 2020, Vietnam’s age-standardized incidence rate per 100,000 cases of GC in men was 21.7, while in women was 10.6. *H. pylori* prevalence rate in Vietnam was very high, at 70.3%. Studies show that Vietnam’s two major cities have a prominent relationship between *H. pylori* infection and non-cardia GC [85], indicating *H. pylori* is also a significant risk factor for GC in Vietnam. Aside from *H. pylori*, tobacco smoking is the second most common risk factor for GC in Vietnam, accounting for 13.5% [86].

### 3.6. Malaysia

Malaysia was one of the few places in Asia with a moderate GC risk [87]. In 2020, the age-standardized incidence rate per 100,000 people was 5.9 for males and 2.8 for females (Figure 2). Although in Malaysia too, men generally are more susceptible to developing GC, the incidence rate of GC is substantially lower than in other Asian countries.

The relatively low incidence rate of GC may be due to a low prevalence rate of *H. pylori*, which stands at 28.6% [69], lower than many other Asian countries. Moreover, ethnicity may also contribute to the low incidence rate. Studies show Malay ethnic people to be less prone to GC than Indians and Chinese [88].

### 3.7. Mongolia

The age-standardized incidence rates of GC in Mongolia were 47.2 per 100,000 people in males and 20.7 per 100,000 in females (Figure 2). Gender differences in age-specific incidence and mortality rates are gradually diminishing in Mongolia [89] but remain to be high. In 2019, the male-to-female ratio of age-specific incidence was 2.34 [89], while in 2020, it slightly decreased to 2.28 (Figure 2).

Like other regions, *H. pylori* is also a dominant risk factor for GC in Mongolia, with an 80% prevalence rate [90]. CagA-positive strains are known to be the most virulent type of GC. However, Mongolia’s high incidence of GC may not apply this strain difference [91].

The age-standardized mortality rate of GC in Mongolia per 100,000 population for both males and females was 25.3 in 2012l it was the highest rate amongst all the other countries. The number of deaths from GC and its associated burden is still increasing in Mongolia [89].

## 4. Challenges for GC Treatment in Asian Countries

Most GC patients are diagnosed in advanced stages where there is low survival rates and few current treatment choices available [92]. This late detection happens as the early symptoms are vague (e.g., weight loss), which are hard to diagnose [93]. Various advances in treatment methods have been rolled out over the years. From the first successful GC surgery in 1881 [94], surgery has been an essential approach to treating GC for years [95]. Other therapies, including immunotherapy, target therapy, and systemic chemotherapy, effectively treat GC [5]. However, despite these treatment options, the 5-year survival rate for advanced GC remains low. Stage I GC’s 5-year survival rate is 68–80%, while for advanced stages, the survival rate drops significantly to below 60% (stage II’s GC: 46–60%; stage III’s GC: 8–30%; and stage IV’s GC: 5%) [96]. This shows that regardless of the advancements in treatment, GC remains a considerable challenge to humankind. The late diagnosis of Asian patients has always resulted in worse prognosis outcomes. Compared to developed countries, the medical resources and treatment choices for GC in developing countries are relatively limited, and early screening in underdeveloped areas is even more inconvenient. Moreover, the large population density in underdeveloped regions also poses challenges to the personalized treatment needs of GC.

The heterogeneity of GC is a nonnegligible obstruction to achieving a better prognosis. Trastuzumab’s success and the rise in GC incidences in younger patients have made discovering targeted therapies for GC increasingly important for overcoming related heterogeneous tumor biology difficulties.

The tumor microenvironment (TME) complexity is a vital factor determining this heterogeneity. TME mediates tumor cells and the surrounding environment, consisting of extracellular matrix (ECM), matrix proteins, cytokines, cancer-associated fibroblasts (CAF), and immune cells. TME plays a crucial role in maintaining the inflammatory environment, ECM remodeling, tumorigenesis, and genetic and epigenetic changes in tumor cells. TME is also important for immune regulation and chemical resistance in various cancers, including GC [97,98]. GC was divided into four subcategories by TCGA, each with a different congenial treatment: Epstein–Barr virus (EBV), microsatellite instability (MSI), chromosome instability (CIN), and genomic stability (GS). ACRG also conducted a transcriptome analysis of 300 samples from the Republic of Korea and reached similar conclusions. However, despite the molecular classification of GC, these insights still need to be translated into clinical practice. Recent works have proved that up to 36% of GC patients exhibit spatial heterogeneity with inconsistencies between primary and paired metastatic focus [99]. Some other research concluded similar findings when comparing tumor samples from the same patient before and after targeted treatment, indicating temporal heterogeneity of GC [100].

The intestinal microbiome’s heterogeneity may highly affect the therapeutic efficacy of GC. The development and prognostic outcomes of GC involve multifarious bacteria, which are mainly present in the oral cavity and gastrointestinal tract. Studies have shown the oncogenic role of several types of bacteria. *Streptococcus anginosus* participates in the metabolism of glucose and lactose to provide energy for GC cell viability maintenance [101]. Fermentation end products such as acetic acid, butyric acid, and isobutyric acid contribute to cell damage and gastric tumorigenesis. Additionally, they were mainly produced by the Gram-positive bacteria *Peptostreptococcus stomatis* and *Slackia exigua* [102]. *Parvimonas micra* was found to be crucial in distinguishing GC from early dysplasia and may be a critical bacterial target for GC treatment [103]. Due to the overall variation in the microbiome in GC patients, it is necessary to investigate whether there are therapeutic benefits to maintaining gastrointestinal homeostasis and develop new strategies to target or reduce these disturbances.

## 5. Forefronts in GC Treatment: Targeted Antibodies and Immunotherapy

The development of new GC therapies mainly focuses on targeted antibody discovery and immunotherapy. Although molecular and cellular evidence suggested that diversified genes and signaling pathways would play influential roles in the occurrence and biological process of GC, only fractional targets are pharmaceutically available. The most successful target for GC treatment is HER2, which is responsible for the transduction of intracellular signals and may induce cell proliferation, migration, and invasion [104]. Research reveals that Trastuzumab can increase overall survival (OS) in HER2-positive cases when combined with standard chemotherapy regimens [105]. Antibody-drug conjugates, such as Trastuzumab deruxtecan (DS-8201) and Disitamab vedotin (RC48), can further enhance the cytotoxicity of HER2 antibodies to counter the acquired drug resistance [106,107]. Angiogenesis is one of the hallmarks of tumorigenesis and is mainly regulated by VEGF/VEGFR signaling axis. Targeted VEGFR has achieved positive results in GC. Despite the barely unsatisfactory progress in discovering VEGF-targeted drugs, Phase III clinical trials have demonstrated that the monotherapy of Ramucirumab and tyrosine kinase inhibitor apatinib can be efficacious in improving OS of GC patients [108,109]. Lenvatinib and regorafenib are well-proved multi-kinase inhibitors with anti-VEGFR activity. They are currently being evaluated in combination with immune checkpoint inhibitors for GC treatment, and inspiring preliminary results have been observed [110,111].

Apart from the canonical therapeutic targets in GC treatment, FGFR2 and Claudin 18.2 have emerged as the new therapeutic targets for GC treatment. Amplification of FGFR2 was observed in up to 15% of GC patients worldwide, and the high expression of FGFR2 is associated with poor prognosis. Bemarituzumab is a monoclonal antibody that can selectively bind to FGFR2b. Hence, it can inhibit the binding of FGFR2 to FGFs and mediate intra-cellular cytotoxicity. A recent study demonstrated that co-administration of bemarituzumab with standard chemotherapy (FOLFOX6) can improve progression-free survival (PFS) and OS of GC patients with highly expressed FGFR2b [112]. The therapeutic effect of combination with immunotherapy is undergoing a phase III clinical trial. Futibatinib is an irreversible and highly selective FGFR1-4 inhibitor under investigation in phase I clinical trials involving patients with advanced solid tumors and FGFR mutation [113]. Zolbetuximab (IMAB362) is a Claudin 18.2-targeted monoclonal antibody. Adding zolbetuximab to chemotherapy can improve PFS and OS with acceptable side effects [16]. A chimeric antigen receptor T cell (CAR-T) therapy targeting claudin18.2 (CT041) was assessed in phase I clinical trial of patients with gastrointestinal cancer. The results presented acceptable safety behavior, a favorable overall response rate (ORR), and an improved 6-month survival rate [114].

Specific candidate targets have presented promising potential in pre-clinic research but have not shown significant improvement in prognosis in clinical trials. EGFR is a well-known oncogenic target in various types of cancer. Researchers have estimated the efficacy of anti-EGFR monoclonal antibodies in a small group of GC patients (19 out of 363 patients) with EGFR amplification. They found that adding cetuximab to chemotherapy in this specific group of patients improved the therapeutic efficacy. However, cetuximab and panitumumab (another monoclonal antibody of EGFR) failed to improve the prognosis of patients with unselected EGFR status [115,116]. Rilotumumab, a monoclonal antibody targeting c-MET, demonstrated anti-tumor efficacy in gastric and esophageal cancers in the phase II trial. However, in the critical phase III RILOMET-1 trial, adding rituximab to chemotherapy failed to improve the outcome of gastric and gastroesophageal cancer. In the VIKTORY trial, vomiting, a selective c-MET tyrosine kinase inhibitor, was administered to patients with MET amplification, leading to favorable response rates and survival [117]. These findings suggest that personalized biomarker molecular diagnosis may offer more clinical treatment options and potentially enhance efficacy.

Immunotherapy has been a significant breakthrough in cancer treatment and has been rapidly advancing over the past decade. Owing to the moderate to high tumor mutation loads in tumors, GC has shown particular sensitivity to immunotherapy, especially in subtypes MSI-H and EBV-related cancers. Phase III clinical trials have demonstrated the efficacy of PD-1 inhibitors, nivolumab and pembrolizumab, in improving the OS in GC patients. In HER2-positive patients, adding pembrolizumab to standard trastuzumab combined chemotherapy reduces tumor size and improves objective response rate. A phase III trial in Asia with 493 patients from Japan, the Republic of Korea, and Taiwan demonstrated that nivolumab had superior benefits over a placebo. Nivolumab brings a median overall survival period of 5.26 months compared to 4.14 months for the placebo group [118]. While PD-1 antibodies have gained popularity in therapeutic applications, the therapeutic potential of PD-L1 is not yet well-understood. Avelumab, an anti-PD-L1 antibody, has failed to improve OS or PFS compared with chemotherapy in the phase III JAVELIN Gastric 300 trial [119]. Meanwhile, the avelumab-treated group did not demonstrate superior overall survival in maintenance therapy compared to continued first-line chemotherapy in all patients or selected PD-L1-positive populations [120].

As summarized in Table 1, despite several confusing results and setbacks, the development of targeted therapies, including monoclonal antibodies, antibody-drug conjugates, small molecule tyrosine kinase inhibitors, and checkpoint inhibitors, have dramatically enriched strategies for GC treatment. Meanwhile, new therapies developed based on molecular biomarkers and signal pathways are promising approaches to promote precise and personalized medical treatment for GC.

## 6. Availability of GC Early Screening, Diagnosis, and Treatments Differs among Asian Countries in an Economy-Based Manner

The classic diagnostic options for GC include endoscopy, biopsy, imaging tests such as computerized tomography (CT) and magnetic resonance imaging (MRI), and laboratory tests such as blood tests [138]. Endoscopy remains the gold standard for diagnosing GC. Endoscopy allows direct visualization of the stomach lining and biopsy of suspicious tissue. A biopsy is essential in determining the type and stage of cancer. Imaging tests are used to assess the extent of cancer spread, including the involvement of lymph nodes and distant metastasis. Laboratory tests evaluate the patient’s overall health and identify underlying conditions that may affect the treatment plan.

There is no significant difference in diagnosis strategies among regions with different levels of economic development. It may be due to the long-term growth of basic medical research in the GC [139]. Classic pathogenesis and universally effective diagnosis and treatment plans have become more prevalent in various countries. In some developed Western countries, the epidemiological incidence rate of GC declined significantly compared to less than a century ago. However, the prevalence of early screening for GC and the availability of medical resources, such as diagnosis and treatment, are positively related to the economic status of regions. Economically developed countries generally have better access to diagnostic options due to higher healthcare spending, better infrastructure, and skilled healthcare personnel. Meanwhile, the cancer burden acceptance varies among patients from areas with different economic development levels [140]. Therefore, it is crucial for local governments to accurately predict the financial burden of emerging chemotherapies and targeted therapies. It can then help the government to develop strategies for stratifying patients according to their potential benefit from these expensive treatments.

GC is difficult to diagnose in the early stages because it usually progresses asymptomatically or only causes non-specific symptoms. The healthcare infrastructure can also delay diagnosis. However, the longer the disease remains undiagnosed, the later the carcinogenesis stage and the poorer the prognosis. Early screening of GC has been identified as a vital strategy in preventing deaths related to the disease. However, access to screening programs and practical diagnostic tools varies across regions and countries with different levels of economic development. The mortality and incidence rates of GC in countries with high health expenditures have been significantly improved in recent years. In Japan and the Republic of Korea, the policy-endorsed popularization programs of early GC screening have improved with increased medical expenses and thus reduced the risk of GC death nationwide [141,142]. The cancer survival rate in China has significantly improved from 2003 to 2015, primarily due to the expansion of medical insurance and the decrease in import taxes on anti-cancer drugs, accompanied by economic development [140]. In economically developed countries, the government will provide more medical subsidies to citizens, especially in the early screening of GC. However, such policy support is relatively rare in developing countries.

From the patient’s perspective, the prognosis of GC is closely related to their socioeconomic status, which may lead to dramatic variation between individuals in both educational levels, emphasis on insurance, acceptance range of cancer burdens, and awareness of GC early screening. Patients with poor economic status may have limited access to diagnostic options due to financial constraints and inadequate healthcare infrastructure. Thus, patients with poor prognosis GI cancers and low socioeconomic status are at substantial risk of undertreatment [143]. A study from India shows that the risk of multiple cancers, including GC, is higher among the uneducated [144]. Another investigation demonstrated that income inequality could significantly affect patients’ acceptance of cancer treatment, even in developed countries. Although universal health insurance in Korea has covered endoscopic mucosal/submucosal resection (EMR) for GC treatment, the lowest income group patients are less likely to receive this treatment [145].

Generally speaking, a better regional economic situation will significantly reduce patients’ barriers to actively participating in GC diagnosis and treatment. Economically developed regions will also have more opportunities to achieve higher early screening prevalence and access to more advanced and personalized treatment programs. Thus, we need to invest more in medical resources in the poor economic regions and populations.

## 7. Conclusions and Future Perspectives

In Asia, both GC incidences and mortalities are declining over time. However, the 5-year survival rate is still meager due to diagnosis in the late stage. Large-scale screening and early diagnosis are crucial to mitigate the risks of GC. Due to the heterogeneity and complex mechanism of GC, the recurrent rates are very high. Combinational administration of small molecules, monoclonal antibodies, and checkpoint inhibitors need to be tested based on molecular features for precision medicine. Additionally, it is still intractable to handle the peritoneal metastasis of GC. More deep mechanism studies on the interplay of GC cells and microenvironments are required to shed light on blocking peritoneal metastasis.

Promoting early screening for GC is crucial to achieving a better prognosis and improving patients’ quality of life. Unfortunately, most Asian patients are typically diagnosed with advanced stages of GC, by which time they have considerable loss of organ function and metastasis of cancer cells. The late diagnosis of GC would contribute to both difficulties in treatment and the high probability of a poor prognosis. Therefore, it is vital to raise awareness about GC screening at a societal level, primarily among groups with an increased incidence of GC. The discovery of molecular mechanisms involved in gastric carcinogenesis could aid in identifying novel biomarkers for early diagnosis or prognostic and treatment–response surveillance. Progress has been made in identifying non-invasive biomarkers from alternative sources, such as saliva, urine, and succus gastricus. A feasible strategy for early diagnosis is to identify circulating molecule-based biomarkers in the early stage of GC. Several types of RNA, including lncRNA, microRNA, and circRNA, have been identified as abnormally expressed in the early stage of GC. RNA detection is quicker, more convenient, and more sensitive than classic GC assessment methods, such as HER2 status and upper gastrointestinal endoscopy. RNA detection can offer better targeted personalized treatment options to deal with the heterogeneity of GC.

To enhance the sensitivity of early screening and broaden the range of early biomarkers, further research should be conducted to optimize laboratory techniques, such as extraction, quantification, probe enrichment, and evaluation methods. Furthermore, independent populations and cohorts should be adopted to validate findings in prospective studies.

## Figures and Tables

**Figure 1 cancers-15-02639-f001:**
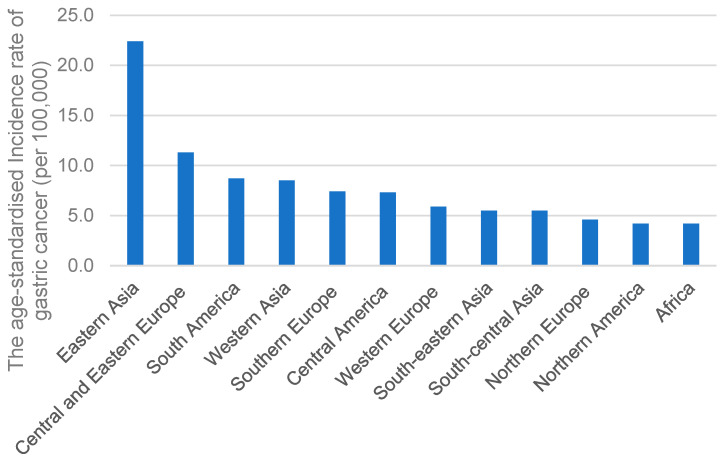
The age-standardized incidence rates of GC in different regions.

**Figure 2 cancers-15-02639-f002:**
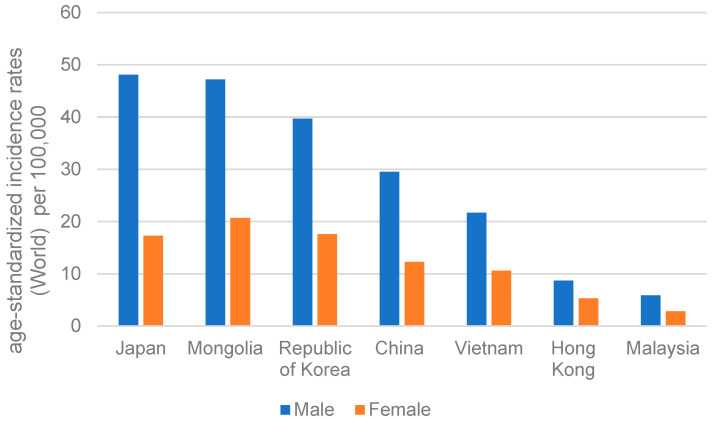
The age-standardized incidence rates of gastric carcinoma for men and women in different countries of Asia in 2020.

**Figure 3 cancers-15-02639-f003:**
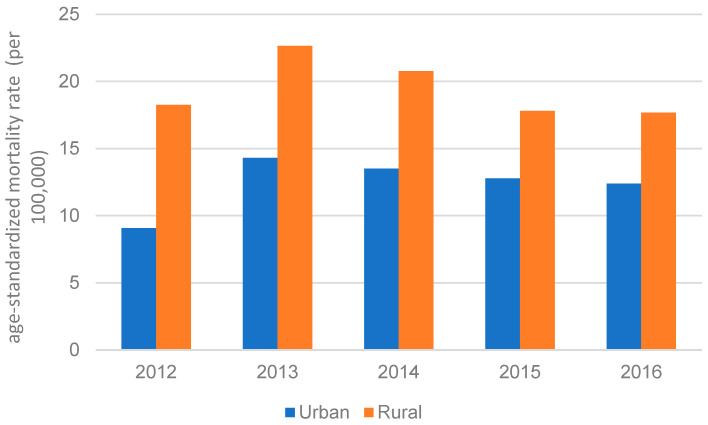
GC age-standardized mortality rate in China between urban and rural areas from 2012 to 2016.

**Figure 4 cancers-15-02639-f004:**
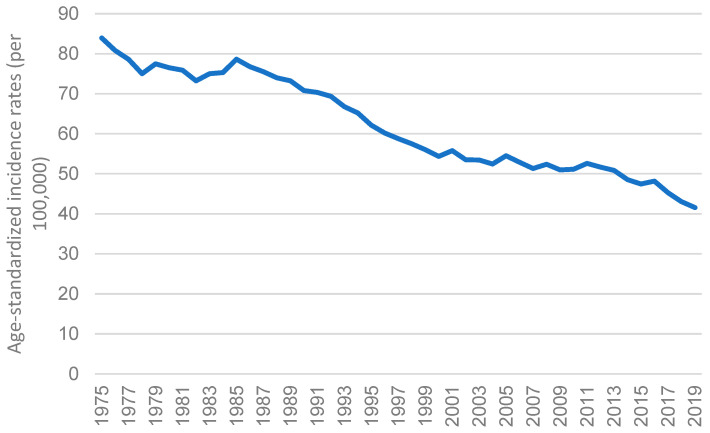
The age-standardized incidence rates of stomach cancer for both sexes from 1975 to 2019 in Japan.

**Figure 5 cancers-15-02639-f005:**
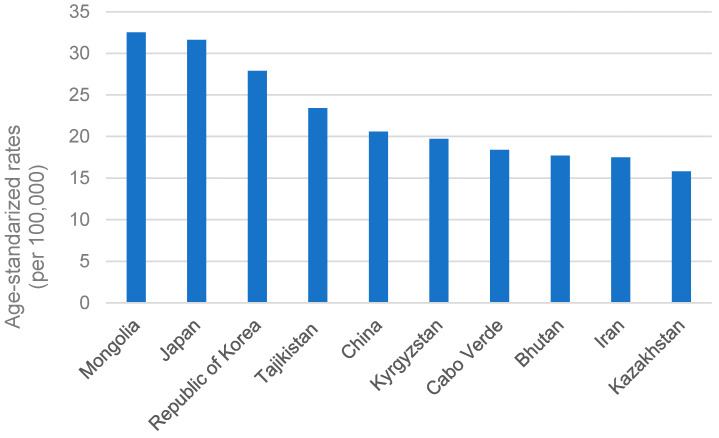
Top 10 age-standardized rates of GC per 100,000 in the leading countries in 2020.

**Figure 6 cancers-15-02639-f006:**
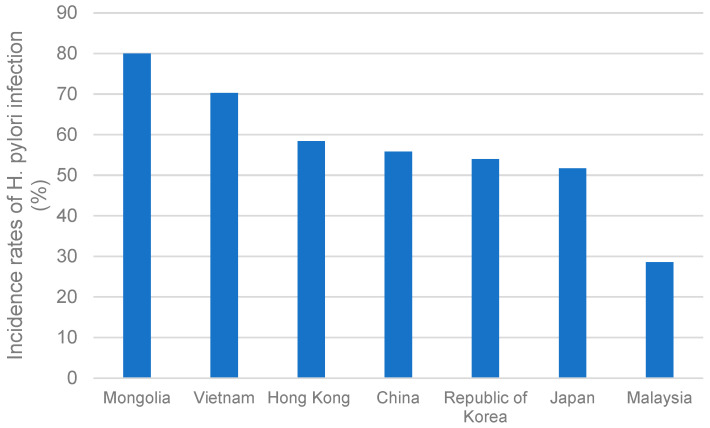
The incidence rates of *H. pylori* infection in different Asia-Pacific regions; China, Japan, Republic of Korea, Vietnam, Malaysia, Hong Kong, and Mongolia.

**Figure 7 cancers-15-02639-f007:**
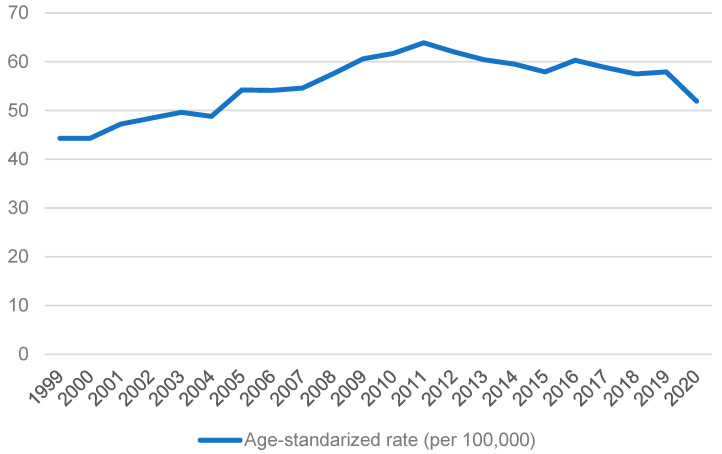
Age-standardized rate of GC from 1999 to 2020 in Republic of Korea.

**Figure 8 cancers-15-02639-f008:**
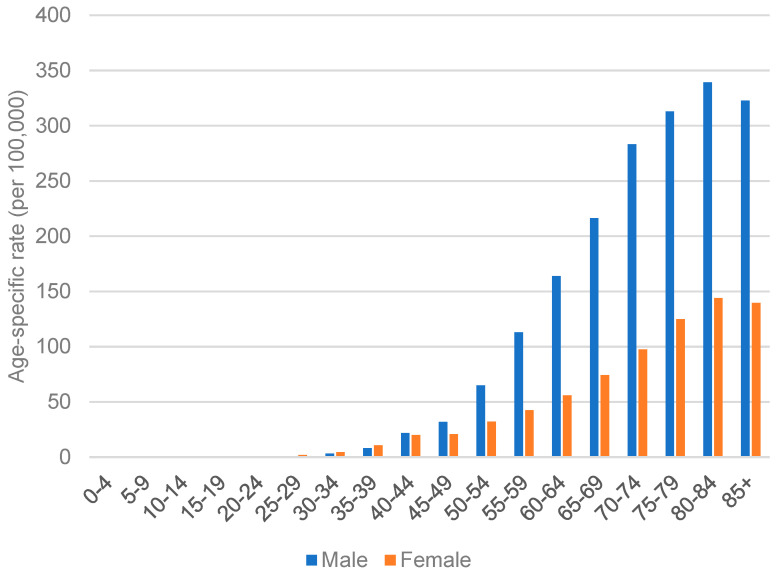
Age-specific rate of GC in both sexes of Republic of Korea in 2020.

**Figure 9 cancers-15-02639-f009:**
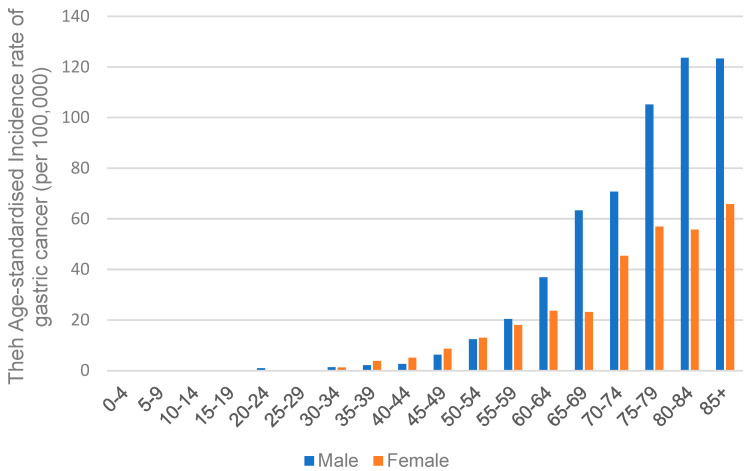
The incidence rates of gastric carcinoma in both sexes among different age groups in Hong Kong.

**Figure 10 cancers-15-02639-f010:**
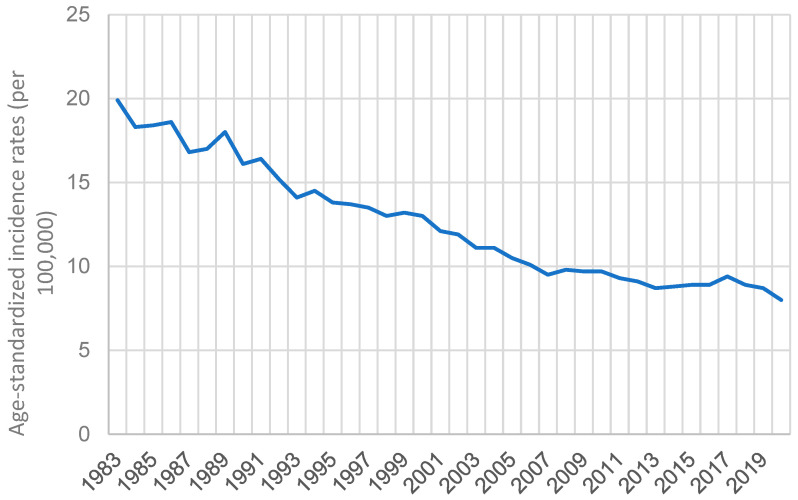
The age-standardized incidence rates (per 100,000) of GC in Hong Kong from 1983 to 2020.

**Table 1 cancers-15-02639-t001:** Recent clinical trials related to GC-targeted therapy.

Target	Drug	Phase	Study	Number of Patients	Outcome	Ref.
HER2	Trastuzumab	III	TOGA	594	Trastuzumab-CP-FP combined group showed improved OS (HR 0.74, *p* < 0.01) and PFS (HR 0.71, *p* < 0.01), with an increased overall response rate (*p* < 0.01) when compared with the chemotherapy group.	[105]
	Lapatinib	III	LOGIC	545	Lapatinib-OXC combined group showed improved PFS (HR 0.82, *p* = 0.038) and PFS (HR 0.82, *p* = 0.038), with increased overall response rate (*p* < 0.01) when compared with the chemotherapy group.	[121]
	Pertuzumab to trastuzumab	III	JACOB	780	No significant improvement in OS (HR 0.84, *p* = 0.057).	[122]
	Trastuzumab emtansine	II/III	GATSBY	345	Trastuzumab emtansine was inferior to taxane in patients with previously treated, HER2-positive advanced GC.	[123]
	Trastuzumab Deruxtecan	II	DESTINY-Gastric01	188	Therapy with Trastuzumab deruxtecan led to significant improvements in response (*p* < 0.01) and OS (HR 0.59, *p* = 0.01) when compared with standard therapies.	[124]
VEGF	Bevacizumab	II	AVAGAST	774	No significant improvement in OS, but increased PFS (HR 0.80, *p* = 0.0037) and overall response rate (*p* = 0.0315).	[125]
VEGFR2	Ramucirumab	III	RAINFALL	645	No significant improvement in OS (HR 0.96, *p* = 0.68) and overall response rate (*p* = 0.17).	[126]
	Ramucirumab	III	RAINBOW	665	The combination of ramucirumab with paclitaxel significantly increases overall survival (HR 0.80, *p* = 0.017) and PFS (HR 0.63. *p* < 0.01).	[127]
	Ramucirumab	III	REGARD	355	Single drug administration of Ramucirumab showed improved OS (HR 0.77, *p* = 0.047) and PFS (HR 0.48, *p* < 0.01).	[108]
c-MET	Rilotumumab	III	RILOMET-1	609	Not effective in improving clinical outcomes in MET-positive GC patients.	[128]
	Onartuzumab	III	METGastric	562	No significant improvement in OS, PFS, and overall response rate.	[129]
EGFR	Panitumumab	III	REAL3	553	Adding panitumumab to EOC chemotherapy is ineffective in improving OS and PFS.	[116]
	Cetuximab	III	EXPAND	904	No additional benefit to combining cetuximab with chemotherapy for advanced GC patients.	[130]
	Gefitinib	III	COG	450	No significant improvement in OS, but it has palliative benefits for patients with short life expectancy.	[131]
FGFR2	AZD4547	II	SHINE	71	AZD4547 did not significantly improve PFS versus paclitaxel in FGFR2 amplification/polysomy GC patients.	[132]
PARP	Olaparib	III	GOLD	643	No significant improvement in OS with olaparib in the overall or ATM-negative population.	[133]
mTOR	Everolimus	III	GRANITE-1	656	No significant improvement in OS (HR 0.90, *p* = 0.124) and overall response rate.	[134]
Claudin18.2	Zolbetuximab	II	FAST	252	Adding zolbetuximab to first-line EOX provides longer PFS (HR 0.44, *p* < 0.001) and OS (HR 0.56, *p* < 0.001) versus EOX.	[16]
	CT041	I	NCT038-74897	123	CT041 has promising efficacy with an acceptable safety profile in CLDN18.2-positive system cancers.	[114]
Immuno-therapy	Pembro-lizumab	III	KEYNOTE-061	650	Single drug administration of Pembrolizumab presented no significant improvement in OS compared to paclitaxel.	[135]
	Pembro-lizumab	III	KEYNOTE-590	383	The combined therapy presented improved OS (HR 0.62, *p* = 0.001) and PFS (HR 0.51, *p* < 0.001) when compared with chemotherapy alone.	[136]
	Nivolumab	III	ATTRAC-TION-02	493	Improved OS (HR 0.63, *p* < 0.001) and PFS (HR 0.60, *p* < 0.001) of patients with advanced gastric or gastro-oesophageal junction cancer.	[118]
	Nivolumab	III	CheckMate-649	955	Adding nivolumab to chemotherapy presented improved OS (HR 0.71, *p* < 0.001, PFS (HR 0.68, *p* < 0.001), and overall response rate (*p* < 0.01).	[137]

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
