# Peer review of "Updated Epidemiology of Gastric Cancer in Asia: Decreased Incidence but Still a Big Challenge"

_cancers, 2023, doi:10.3390/cancers15092639_

Round 1

Reviewer 1 Report

 This manuscript gives a global view about the gastric cancer classification, prevalence, risk factors, and treatment in Asia. The authors need to improve their writing style. Some paragraphs are difficult to understand, e.g., lines 91, 210-212, 250-252, 266, 303 (have to be “H. pylori”), 356, 432, 439 and others. The manuscript needs to be checked by native English speakers.  Some references are not corresponding to the text, e.g. Ref 52 is about Curcumin and not signaling pathways triggered by smoking. They have to be positive that This manuscript gives a global view about the gastric cancer classification, prevalence, risk factors, and treatment in Asia. The authors need to improve their writing style. Some paragraphs are difficult to understand, e.g., lines 91, 210-212, 250-252, 266, 303 (have to be “H. pylori”), 356, 432, 439 and others. The manuscript needs to be checked by native English speakers.  Some references are not corresponding to the text, e.g. Ref 52 is about Curcumin and not signaling pathways triggered by smoking. They have to be positive that all references are correctly cited. The phrase cited in line 203 came from a review (Ref 49) where was cited too. Authors have to cite the original paper.

all references are correctly cited. The phrase cited in line 203 came from a review (Ref 49) where was cited too. Authors have to cite the original paper.

The manuscript require an extensive editing of English language. A reexamination of the whole manuscript by a native English speaker will be helpful.

Author Response

Point-by-point Response:

Response to Reviewer #1:

Q1: This manuscript gives a global view about the gastric cancer classification, prevalence, risk factors, and treatment in Asia. The authors need to improve their writing style. Some paragraphs are difficult to understand, e.g., lines 91, 210-212, 250-252, 266, 303 (have to be “H. pylori”), 356, 432, 439 and others. The manuscript needs to be checked by native English speakers. 

Response: We sincerely thank you for the positive comments and suggestions about our work. Based on your comments, we made some corrections to the manuscript to make the content more fluent and understandable. We also invited a native English speaker to double-check the manuscript thoroughly.

    Specifically, on lines 91, 210-212, 250-252, 266, 356, 432, and 439, we have edited and rephrased the corresponding sentence to increase its understandability. On line 303, we have changed the word to H. pylori to make the manuscript more coherent.

Q2: Some references are not corresponding to the text, e.g. Ref 52 is about Curcumin and not signaling pathways triggered by smoking. They have to be positive that all references are correctly cited.

Response:  Thanks for pointing out this issue, and we agree with your comments. We have edited this section regarding the points you raised and included more relevant References, namely Ref 52 “Machlowska, Julita, Jacek Baj, Monika Sitarz, Ryszard Maciejewski, and Robert Sitarz. "Gastric Cancer: Epidemiology, Risk Factors, Classification, Genomic Characteristics and Treatment Strategies." International Journal of Molecular Sciences 21, no. 11 (2020): 4012”, and Ref 53 “Jensen, Kendal, Syeda Afroze, Md Kamruzzaman Munshi, Micheleine Guerrier, and Shannon S Glaser. "Mechanisms for Nicotine in the Development and Progression of Gastrointestinal Cancers." Translational gastrointestinal cancer 1, no. 1 (2012): 81.”, to replace the original Ref 52.

Q3: All references are correctly cited. The phrase cited in line 203 came from a review (Ref 49) where was cited too. Authors have to cite the original paper.

Response: We genuinely thank you for the suggestive comments on our citations and suggestions. Based on the suggestion, we have revised and cited the original paper you mentioned.

Q4: The manuscript require an extensive editing of English language. A reexamination of the whole manuscript by a native English speaker will be helpful.

Response: Thanks a lot for pointing out this issue. We have invited a native English speaker to correct the manuscript thoroughly and hope it reaches the publication standard.

Reviewer 2 Report

Current topic

Comprehensive literature review

Comprehensive epidemiological evaluation

Appropriate title and introduction

I have not found references to the methods of diagnosis and therapeutic possibilities. I believe that it is necessary to specify the diagnostic and therapeutic pathways in the various Asian countries. Are there differences in different countries? Have more economically developed countries adopted diagnostic and treatment options that are difficult for more economically poor patients?

Author Response

Response to Reviewer #2:

Q1: I have not found references to the methods of diagnosis and therapeutic possibilities. I believe that it is necessary to specify the diagnostic and therapeutic pathways in various Asian countries. Are there differences in different countries? Have more economically developed countries adopted diagnostic and treatment options that are difficult for more economically poor patients?

Response: Thanks a lot for providing such an important issue. Yes, we totally agree with your suggestions. We have updated the summary Table for ongoing GC therapeutic trials with corresponding References. Moreover, we also added a new section to discuss the discrepancies in early screening, diagnosis, and treatments of GC between countries with different economic statuses on pages 17-18. We hope the related modifications can address your valuable concerns and improve this manuscript's quality.

Round 2

Reviewer 1 Report

The text that the authors reviewed and modified following my suggestions has improved significantly. Nevertheless, although the text is likely to be understood by a reader it is not easy to read as the readability score of the manuscript (Flesch reading-ease test) is well below the 40 points score and normally a clear, concise and easy to read, understand and follow text requires at least a 70 points score. That is why I recommend  moderate editing of english language. Maybe the authors should contemplate the use of specific readability evaluation programs.

The text that the authors reviewed and modified following my suggestions has improved significantly. Nevertheless, although the text is likely to be understood by a reader it is not easy to read as the readability score of the manuscript (Flesch reading-ease test) is well below the 40 points score and normally a clear, concise and easy to read, understand and follow text requires at least a 70 points score. That is why I recommend  moderate editing of english language. Maybe the authors should contemplate the use of specific readability evaluation programs.

Author Response

Response to Reviewer #1:

Q1: The text that the authors reviewed and modified following my suggestions has improved significantly. Nevertheless, although the text is likely to be understood by a reader it is not easy to read as the readability score of the manuscript (Flesch reading-ease test) is well below the 40 points score and normally a clear, concise and easy to read, understand and follow text requires at least a 70 points score. That is why I recommend  moderate editing of english language. Maybe the authors should contemplate the use of specific readability evaluation programs.

Response: We genuinely thank you for the positive comments and suggestions about our work. We have learned a lot. Based on your comments, we have incorporated specific readability evaluation programs and made moderate editing to the English language for the whole manuscript (the overall score is 96 by the Grammarly report). The passive sentence percentage is lowered from 14.6% to 9%, which is generally acceptable (below 10%) and improves our manuscript's readability. Most complex sentences are separated into two or more simple sentences for better readability. We have tried our best to modify the expression pattern and enhance the readability. Now, the Flesch reading-ease test score you mentioned was improved to 36% (by Grammarly), which is within the range that college students can understand. 

Reviewer 2 Report

No further comment

Author Response

We sincerely appreciate your time and endeavor in the review of this manuscript for two rounds. Moreover, we have learned a lot from your comments. Thank you.